# Preparation of Nanocellulose Aerogel from the Poplar (*Populus tomentosa*) Catkin Fiber

**Yan Wu** [1,2], **Meng Sun** [1,2], **Xinyu Wu** [1,2], **Tianlin Shi** [1,3], **Hong Chen** [1,2] and **Hankun Wang** [3,4,*]

[1] College of Furnishings and Industrial Design, Nanjing Forestry University, Nanjing 210037, China
[2] Co-Innovation Center of Efficient Processing and Utilization of Forest Resources, Nanjing Forestry University, Nanjing 210037, China
[3] Department of Biomaterials, International Center for Bamboo and Rattan, Beijing 100102, China
[4] DepaSFA and Beijing Co-built Key Laboratory of Bamboo and Rattan Science & Technology, State Forestry Administration, Beijing 100102, China
[*] Correspondence: wanghankun@icbr.ac.cn

**Abstract:** The effects of chemical pretreatment on the purification of poplar (*Populus tomentosa*) catkin fiber and the effect of ultrasonic time for the microfibrillarization of poplar catkin fiber (PCF) were studied. The nanocellulose aerogels were prepared by freeze drying the cellulose solutions. The density, porosity, micro morphology, thermal stability and mechanical properties of the aerogels were analyzed. It was found that the dewaxing time of PCF is shorter than that of unsonicated nanocellulose. After the treatment of 0.5 wt% sodium chlorite for 2 h, the lignin of PCF was removed. After the chemical purification, the PCF was treated with 2 and 5 wt% NaOH solution and ultrasonicated for 5 and 10 min, respectively. When the ultrasonic time was 10 min, the diameter of the nanocellulose was 20–25 nm. When the ultrasonic time was 5 min, the aerogels with porous honeycomb structure can be prepared by using the nanocellulose sol of PCF as raw material. The density of the aerogels was only 0.3–0.4 mg/cm$^3$ and the porosities of the aerogels were all larger than 99%. The difference between the pyrolysis temperature of aerogels was small, the elastic modulus of aerogels was 30–52 kPa, and the compressive strength was 22–27 kPa. With the increase of the concentration of NaOH solution (5 wt%) and ultrasonic time (10 min), the elastic modulus of aerogels increased gradually and reached the maximum value of 52 kPa, while the compressive strength reached the maximum value of 27 kPa when the PCF being treated in 5 wt% NaOH solution and was ultrasonicated for 5 min.

**Keywords:** poplar catkin fiber; nanocellulose; aerogel

## 1. Introduction

Poplar is a fast-growing tree species with short growth cycle and low price that is planted in a wide area mainly concentrated in the northeastern, northern and northwestern regions in China. The seed tassel fiber is a kind of plant fiber, the main composition are cellulose and hemicellulose. The lignin content is low and the wall is thin and hollow [1]. The seed tassel fiber has light weight, so it is easy to float and is hard to collect, which is harmful to the environment [2]. Take poplar seed tassel fiber (metamorphosis pericarp of poplar, commonly known as 'Poplar (*Populus tomentosa*) catkin fiber (PCF)') as an example, every year from April to June, the flying PCF causes great trouble for people's lives, not only affecting air quality, but also reducing the cleanliness and visibility of urban roads, and even cause sensitive people allergic, further irritating the skin and respiratory diseases. Therefore, most scholars are concerned about how to suppress the production of PCF but ignore how to use them. In order to utilize the PCF, it is necessary to understand the basic properties of it. The

microstructure and chemical composition of the PCF make a large difference to its properties [1]. Thus studying the structure and chemical composition of PCF may play a vital role in the use of PCF in high value-added fields such as cellulose.

Cellulose exists in the cell wall of plants in the form of microfibrils with a diameter of 3–5 nm. Microfibrils can aggregate to form cellulose aggregates with a diameter of several tens of nanometers and a length of several tens of micrometers, which is called cellulose nanofibers (CNFs). Compared to the microfibrillated cellulose and nanocellulose whiskers, its unique supramolecular structure and morphology make it possess excellent mechanical, optical and small molecular physical barrier properties [3]. It has broad application prospects in new biomaterials, medicine, information and other industrial fields [4]. At present, the source of nanocellulose is mainly from wood, but due to the long growth cycle and wide use of wood, it is urgent to find more suitable raw materials for high value-added nanocellulose. The scholars of related field have done a lot of researches to prepare the nanocellulose by using the cheap raw materials such as bamboo [5,6], tobacco stalk [7], cotton pulp [8–10], sisal fibers [11], crop straw [12,13], peanut shell [14], soybean hulls [15], banana peel and rod [16], coconut shell [17], cactus peel [18] and beet root [19].

CNF aerogel (CNFA) is a natural polymer aerogel material. It not only has the advantages of traditional aerogel, but also has the characteristics of natural materials which are pollution-free and biodegradable. The preparation of CNFA is mainly divided into two parts: the first is the preparation of the CNF sol, and the second is the drying of the sol. The primary cause of unique structure of the CNFA is that the CNFs with high aspect ratio will cross linking with hydrogen bond and the bond energy between the molecular chains of cellulose causes a unique three-dimensional (3D) network structure when dispersed in an aqueous solution. After freeze drying, CNFA can still retain this kind of 3D structure. Studies have shown that CNFA has better mechanical properties. Cai et al. [20] obtained cellulose hydrogel after treating cellulose with LiOH and urea solution, and then prepared the cellulose composite aerogel by supercritical drying, which has high transparency and excellent mechanical properties. The compression modulus is 200–300 MPa, which is two orders higher than that of traditional inorganic aerogels. Sehaqui et al. [21] prepared cellulose fibrils with high aspect ratio (MFC) from wood pulp and obtained aerogel with a porosity of 93.1% by freeze-drying method, which possessed excellent mechanical properties compared with other aerogels. Han et al. [22] prepared CNFs with the average diameter of 15 nm by chemical pretreatment combined with high-frequency ultrasonic method, using bamboo as raw material and prepared cellulose composite aerogel by freeze-drying. Liu et al. [23] used CNF solution as a raw material and added chloroacetic acid (CAA) and aminopropyl triethoxysilane (APTES) as crosslinking agents to prepare CNF aerogel which had the density of 7.55 mg/cm$^3$, the specific surface area of 9.35 m$^2$/g, the porosity of 99.0% and good water absorption and recyclability. Xiao et al. [24] used pine needle as raw material to prepare nanocellulose hydrogel by combining chemical and ultrasonic treatments and then obtained aerogel by freeze-drying. The results showed that the thermal decomposition temperature of the obtained aerogel increased by 56 °C. PCF, the seed fiber of poplar, which is mainly composed of cellulose, hemicellulose and lignin, has objective conditions for preparing nanocellulose. The average yield of poplar fiber per poplar tree is 25 kg and the total area of poplar forest in China have reached 10 million hm$^2$ [25]. Therefore, if the collection method is appropriate, the amount of resources will be considerable. The large amount of PCF produced every year can not only reduce the environmental pollution but also produced obvious economic benefits, since it can turn waste into treasure. PCF was used as the original material in this experiment, because of its high content of cellulose and low content of hemicellulose and lignin. The content of holocellulose of PCF is up to 96%, in which the content of celllose is 44.59% and hemicellulose is 52.32%. The content of ligin is 2.92%. This was tested based on GB/T 2677.10, GB/T 2677.8 and GB/T 744. The hemicellulose and lignin were removed by chloroform dewaxing and chemical pretreatment, and the cellulose nanofibrils (CNFs) were prepared by ultrasonic treatment. The effect of different concentrations of NaOH on the extraction of CNFs and the effect of different ultrasonic treatment time for the micro fibrinolysis of PCF was studied. The CNF Aerogel (CNFA)

was prepared by freeze drying and evaluated by a Scanning Electron Microscope (SEM), Atomic Force Microscope (AFM), Thermogravimetric (TG), FourierTransform Infrared Spectrometer (FTIR) and compression tests.

## 2. Preparation of PCF Nanocellulose

### 2.1. Experimental Materials

PCF, which was produced in Beijing, China; 80% sodium chlorite; chloroform, glacial acetic acid, potassium hydroxide and ethanol, which were all analytical pure.

### 2.2. Experimental Methods

#### 2.2.1. Chemical Purification of PCF

2 g PCF was put into 100 mL chloroform beaker, stirred strongly in 70 °C constant temperature water bath (HH-4) for 20, 40, 60 and 80 min separately, and washed with deionized water combined with pressure filter. This step was repeated three times and the dewaxing process was finished. Then milled the dewaxed PCF into about 1 mm long fiber by using Weifao mill 60XC which produced by Tianjin Huike instrument equipment Co., Ltd (Tianjin, China). Because PCF floats easily during milling, it was first frozen in liquid nitrogen and then ground. The lignin was removed by 1% sodium chlorite solution at 75 °C (the pH of PCF was adjust to 4–5 with glacial acetic acid) [26], and washed to white color by deionized water combining with air pressure filter (TYW-2), this process was repeated three times to achieve cellulose. Finally, the total cellulose was treated with 2 wt% and 5 wt% sodium hydroxide (NaOH), respectively, in 90 °C water bath to remove hemicellulose.

#### 2.2.2. Preparation of PCF Nanocellulose

The purified PCF sample was mixed into 1 wt% solution by water analyzer, and treated into PCF nanocellulose with a high frequency sonication instrument (JY99-IIDN) which was provided by Ningbo Xinzhi Biotechnology Company (Ningbo, China). The basic parameters of the instrument in the experiment were designed as maximum power of 1800 W, an output power of 30%, an ultrasonic frequency of 19.5–20.5 kHz, a probe diameter of 20 mm, a pulse treatment of 5 s, a stop of 5 s, and the ultrasonic time of 5 and 10 min, respectively.

#### 2.2.3. Preparation of PCF Nanocellulose Aerogel

The PCF nanocellulose aerogels (PCFAs) were prepared by freeze drying the nanocellulose sol. The effects of NaOH concentration and ultrasonic time on the aerogels were investigated. The PCF nanocellulose sols were available within the small bottle of 5 mL (in order to reduce the error of mechanical tests, each vial was filled with the same quality sol) and kept in the −4 °C refrigerator precooling for 24 h, then the frozen sols was took out, and frozen in the liquid nitrogen to make its upright in liquid nitrogen for 10 min. Finally, the samples were placed in the freeze dryer to vacuum freeze drying to obtain the nanocellulose aerogels.

#### 2.2.4. Characterization of Nanometer Cellulose from PCF

Fourier Transform Infrared Spectrometer (FTIR) Test

The qualitative characterization of chemical composition changes of PCFAs was carried out by the FTIR (Avance 300, Bruker Company, Berlin, Germany). After the sample was fully dried, it was ground into powder according to the ratio of 1:100 to potassium bromide, and KBr tablet method was adopted for testing. The transmission mode of infrared microscope was selected when testing, the frequency of spectrum scanning was 200 times, and the range of spectrum acquisition was 4000–400 cm$^{-1}$.

Micromorphological Test

The phase imaging mode of atomic force microscope (Icon) of Bruker Company (Berlin, Germany) was used to characterize the changes of cellulose morphology and size during the preparation of PCFAs. At room temperature, the Berkovich nanoprobe with a radius of 8 nm was used with the vibration frequency of 320 kHz, the scanning speed of 1 Hz, the force constant of 42 N/m and the micro cantilever of 125 nm.

### 2.2.5. Characterization of PCF Nanocellulose Aerogels

Density and Porosity

The density of nanocellulose aerogels is obtained by the ratio of mass to volume. The diameter and height of nanocellulose aerogels are measured by Vernier calipers and then calculated.

The porosity of nanocellulose aerogels is calculated from the following formula:

$$P = (1 - \frac{\rho_1}{\rho_2}) \times 100\%$$

where $P$ is the porosity of aerogels, $\rho_1$ is the density of nanocellulose aerogels and $\rho_2$ is 1.5 g/cm$^3$, which is the density of cellulose [27].

Microstructure of Aerogels

In this study, envrionmental scanning electron microscope (Philips Fei XL30 ESEM-FEGG, Amsterdam, Holland) was used to characterize the aerogels prepared under different conditions. The aerogel was frozen in liquid nitrogen, then cut off the cross section to put on the carrier with conductive adhesive and then sprayed it with gold to prepare the samples.

Thermal Stability of Aerogels

The thermal stability of aerogels prepared under different conditions was characterized by thermogravimetric analyzer (TGA) (Type 5500) of Navas Company, CA, USA. The heating rate was 20 °C/min and the temperature range was 30–600 °C.

Mechanical Properties

The mechanical properties of aerogels were characterized by 5848 micro mechanical testing machine made by Instron Company (NY, USA). Each group of 10 samples has a diameter of about 12 mm, a height of about 13 mm, a compression deformation of 90% and a compression rate of 5 mm/min.

## 3. Results and Discussion

### 3.1. Effect of Sonication Time and Chloroform Treatment Time on Dewaxing Effect

Figure 1 shows the infrared spectrum in the range of 500–4000 cm$^{-1}$ and 1000–1800 cm$^{-1}$ of the processing time of the sonication and chloroform treatment to the PCF after dewaxing treatment. From top to bottom are the control samples of PCF, the samples with chloroform dewaxing treatment without sonication for 20, 40, 60 and 80 min and with chloroform treatment after sonication for 20 min, respectively. The wax layer was mainly composed of long chain fatty acids, acyl groups and esters. The characteristic peak at 2918 cm$^{-1}$ was caused by the vibration of CH stretching aliphatic group (=CH$_2$ and CH), and the characteristic peak at 1738 cm$^{-1}$ was generated by the vibration of carboxyl group and C-O stretching acetyl group. As shown in Figure 1, the two characteristic peaks of chloroform treatment without sonication for 20 min of PCF did not change obviously. The results showed that the waxy layer on the surface of PCF could not be removed by chloroform treatment

for 20 min, but after 40 min of chloroform dewaxing, the characteristic peaks at 2918 cm$^{-1}$ and 1738 cm$^{-1}$ were obviously weakened. When the time increased, the change was not obvious. At this time, the diffraction peaks at 2918 cm$^{-1}$ and 1738 cm$^{-1}$ may be cellulose-CH stretching vibration peak and xylan C=O stretching vibration peak, while the chloroform treatment can only remove the aliphatic and acyl groups in the waxy layer, but cannot remove the cellulose and xylan. It can be seen from the Figure 1 that the characteristic peaks at 2918 cm$^{-1}$ and 1738 cm$^{-1}$ of PCF dewaxing after sonication for 20 min were obviously weakened, and the effect can basically reach the effect of dewaxing treatment without sonication for 20, 40, 60 and 80 min of PCF. This is because the length to diameter ratio of PCF is relatively large, it is easy to be entangled and agglomerated in solution. However, the sonication can shorten the length of PCF and it can be fully exposed to the chloroform solution, in this case, the dewaxing effect would be better and the amount of chemical reagent would be greatly reduced.

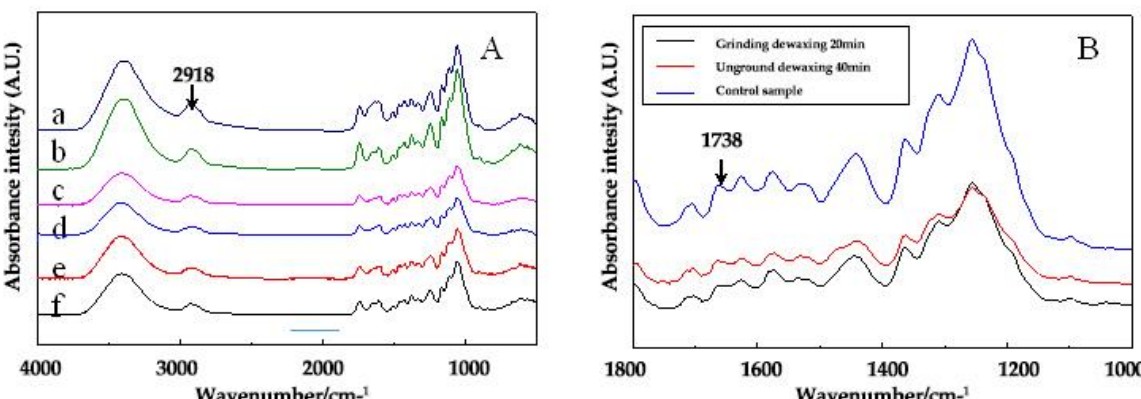

**Figure 1.** The infrared spectrum and the local magnified infrared spectrum after the processing time of the sonication and chloroform treatment to the poplar catkin fiber (PCF) after dewaxing treatment ((**A**) is the total infrared spectrum, (**B**) is the local large infrared spectrum a is the control sample, b–e are only dewaxing without sonication for 20, 40, 60, 80 min, f is sonication and dewaxing for 20 min).

Figure 2 shows the infrared spectra of PCF in the range of 500–4000 cm$^{-1}$ and 1000–1800 cm$^{-1}$ during delignification. It can be seen from Figure 2 that compared with the control sample, the characteristic peaks generated by the stretching vibration of benzene ring at 1423 cm$^{-1}$ and 1510 cm$^{-1}$ changed obviously after the treatment of delignification with sodium chlorite solution of the PCF [28], especially the characteristic peaks of 1510 cm$^{-1}$ almost disappeared. The lignin content of PCF was only 2.92%, which indicated that after treatment with 0.5 wt% sodium chlorite for 2 h, the lignin of the PCF could be basically removed. While bamboo and wood, as raw materials commonly used in the preparation of nanocellulose, have a lignin content of 16–34%, which need to be treated with five times of 1 wt% sodium chlorite solution for 5 h before the lignin can be removed. Therefore, compared with raw materials such as wood and bamboo, the nanocellulose prepared by PCF can greatly reduce the amount of chemical reagents and the processing time.

The characteristic peaks of hemicellulose were mainly at 805 cm$^{-1}$, 1600 cm$^{-1}$ and 1730 cm$^{-1}$ of which 805 cm$^{-1}$ is mainly the characteristic diffraction peak of mannan-ring skeleton vibration. 1600 cm$^{-1}$ and 1730 cm$^{-1}$ were mainly the characteristic peaks of C=O stretching vibration in xylan [29]. It can be seen from Figure 3, compared with the contrast, after 2 wt% NaOH treatment for 2 h and 4 h, the characteristic peaks of C=O stretching vibration in Xylan disappeared in the infrared spectrum at 1730 cm$^{-1}$, while the characteristic peaks generated by C=O stretching vibration of in xylan at 1600 cm$^{-1}$ had no obvious change. After treatment with 5 wt% NaOH for 2 h, the characteristic peaks generated by C=O stretching vibration in xylan at 1730 cm$^{-1}$ also disappeared, while the characteristic peaks at 1600 cm$^{-1}$ continued to exist, but the infrared spectrum was slightly lower than that of 2 wt% NaOH for 2 h or 4 h. The decreasing trend of characteristic peaked at 1600 cm$^{-1}$ after 4 h treatment with

5 wt% NaOH was more obvious, which indicated that 5 wt% NaOH treatment for 4 h is more effective than 5 wt% NaOH treatment for 2 h and 2 wt% NaOH solution for 2 h and 4 h. It could remove a large amount of hemicellulose from PCF; however, a part of hemicellulose still existed. However, a small amount of hemicellulose is helpful to the microfibrillation of purified cellulose [30].

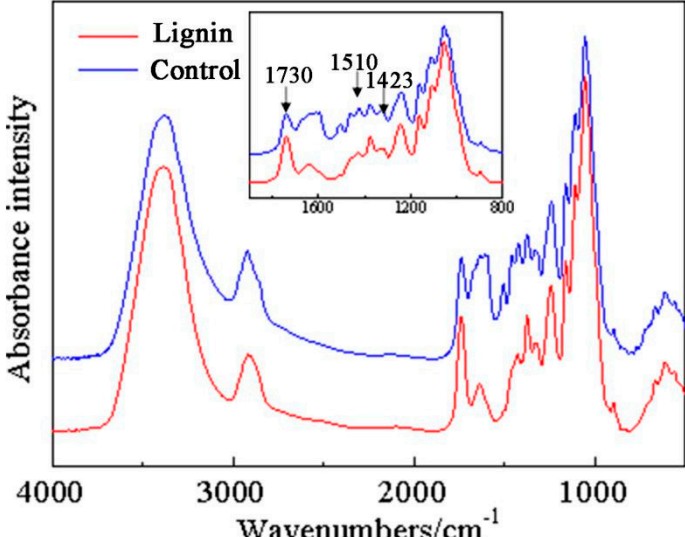

**Figure 2.** Infrared spectra and local amplified infrared spectra of PCF during chemical purification.

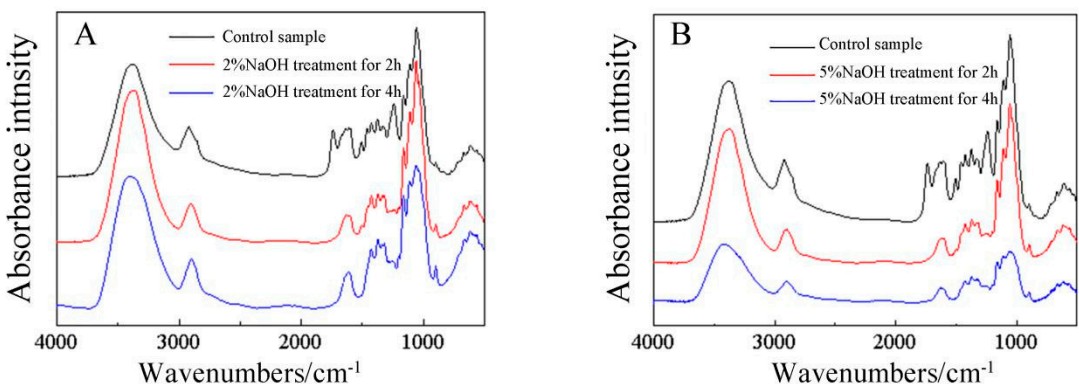

**Figure 3.** Infrared spectra of the effect of different NaOH concentration and time on the hemicellulose removal of PCF. (**A**): control sample, 2% NaOH treatment for 2 h and 2% NaOH treatment for 4 h, (**B**): control sample, 5% NaOH treatment for 2 h and 5% NaOH treatment for 4 h.

### 3.2. Effects of Different Treatment Conditions on the Size and Morphology of Nanocellulose

In order to obtain the morphology and size of PCF nanocellulose accurately, AFM was used to observe the nanocellulose treated under different conditions. Figure 4 gives the preparation of cellulose solution and nanocellulose sol from PCF and Figure 5 shows the microscopic appearance of PCF nanocellulose treated with different concentration of NaOH and different ultrasonic time.

The cellulose obtained from the chemical purification of PCF can be rapidly microfibrillated after being treated with a high-frequency sonication apparatus for only 5 min, and the sol with good morphology can be obtained. If the ultrasonic time continues to increase, the aggregation of PCF in the sol will increase due to the fact that sonication can evenly disperse the fiber of PCF for a certain time. If the time is too long, the fiber of PCF will be entangled and agglomerated. Therefore, this experiment studied the two-phase PCF of the ultrasonic time of 5 min and 10 min. The microscopic morphology of the sol was shown in Figure 5. It can be seen that the sols under different conditions can obtain the nanocellulose with good appearance and uniform dispersion after

ultrasonic treatment for 5 min. And the size of nanocellulose was at the nano-level. Compared with the preparation of nanocellulose from bamboo and wood, which require high pressure homogenization for 10–15 times or need sonication for 30 min, the energy consumption of preparing nanocellulose from PCF was greatly reduced.

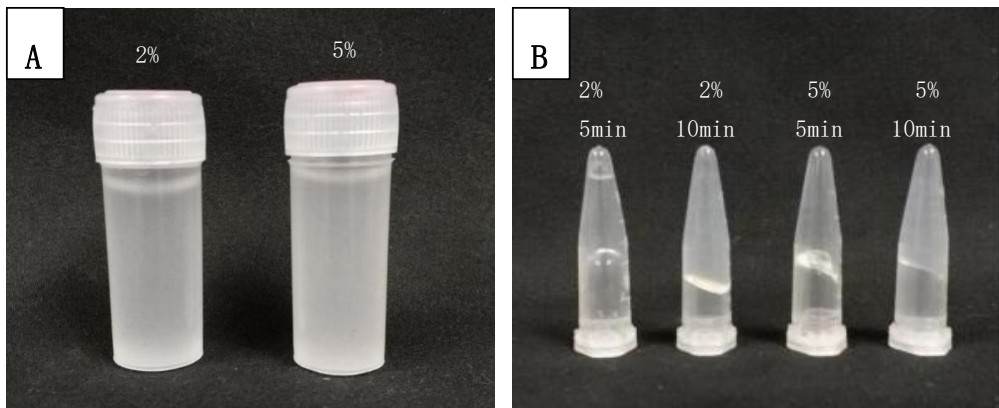

**Figure 4.** Preparation of cellulose solution (**A**: 2 and 5 wt% NaOH treated samples) and nanocellulose sol (**B**: 2 wt% NaOH and 5 min sonication; 2 wt% NaOH and 10 min sonication; 5 wt% NaOH and 5 min sonication; 5 wt% NaOH and 10 min sonication) from PCF.

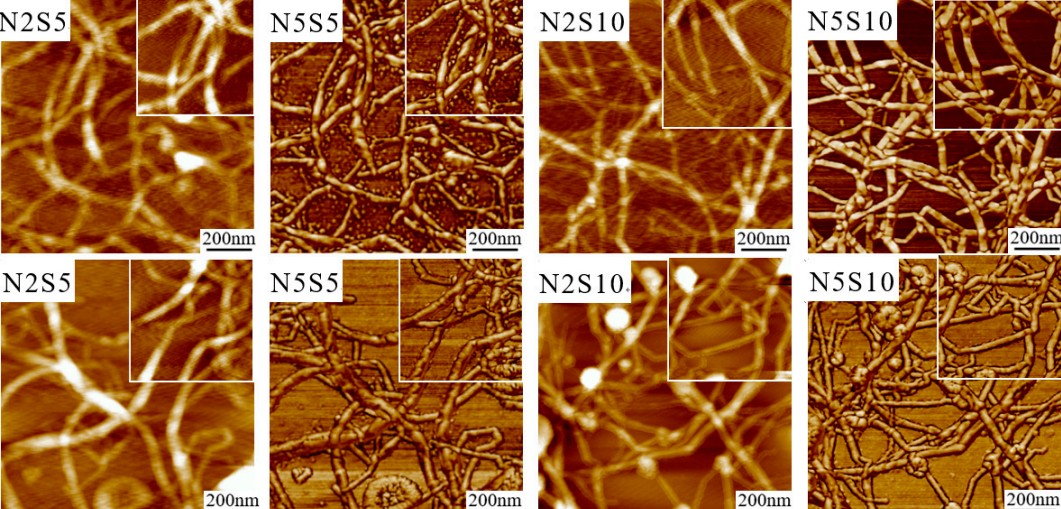

**Figure 5.** The morphology of PCF cellulose after 2 and 5wt% NaOH treatment with atomic force microscope at different time. (N2S5: 2 wt% NaOH and 5 min sonication, N5S5: 5 wt% NaOH and 5 min sonication N2S10: 2 wt% NaOH and 10 min sonication, N5S10: 5 wt% NaOH and 10 min sonication N2S5 and N2S10 for height map, N5S5 and N5S10 for phase map).

As shown in Figure 6, in order to further study the effect of different NaOH concentration and different ultrasonic time on the size of nanofibers prepared by PCF, the size of nanosometers obtained from four different conditions was measured in this experiment, and 100 data were measured in each group, with a total of 400 data.

It can be seen from Figure 6 that the diameter of nanocellulose of PCF is in the range of 15–70 nm. Figure 6A showed the size distribution of N2S5. The size distribution of nanocellulose was in the range of 20–60 nm. The largest size was 35–40 nm, accounting for 38%, and the second was between 40–45 nm, accounting for 22%. Figure 6B showed the size distribution of N2S10. The largest size was in the range of 25–30 nm, which made up for 34%, followed by the size of 40–45 nm, accounting for 24%, indicating that the ultrasonic time is prolonged and the size of nanocellulose is reduced. Figure 6C

shows the size distribution of N5S5. The size distribution of nanocellulose was the most, which was at 25–30 nm accounting for 32%, followed by 35-40 nm, accounting for 22%, and Figure 6D showed the size distribution of N5S10. The nanocellulose size at 20–25 nm was the most, accounting for 32%, followed by 3–35 nm, accounting for 22%. Compared with N2S10, nanocellulose was smaller in size and its distribution was relatively uniform. Therefore, N5S5 was the smallest and the most uniform.

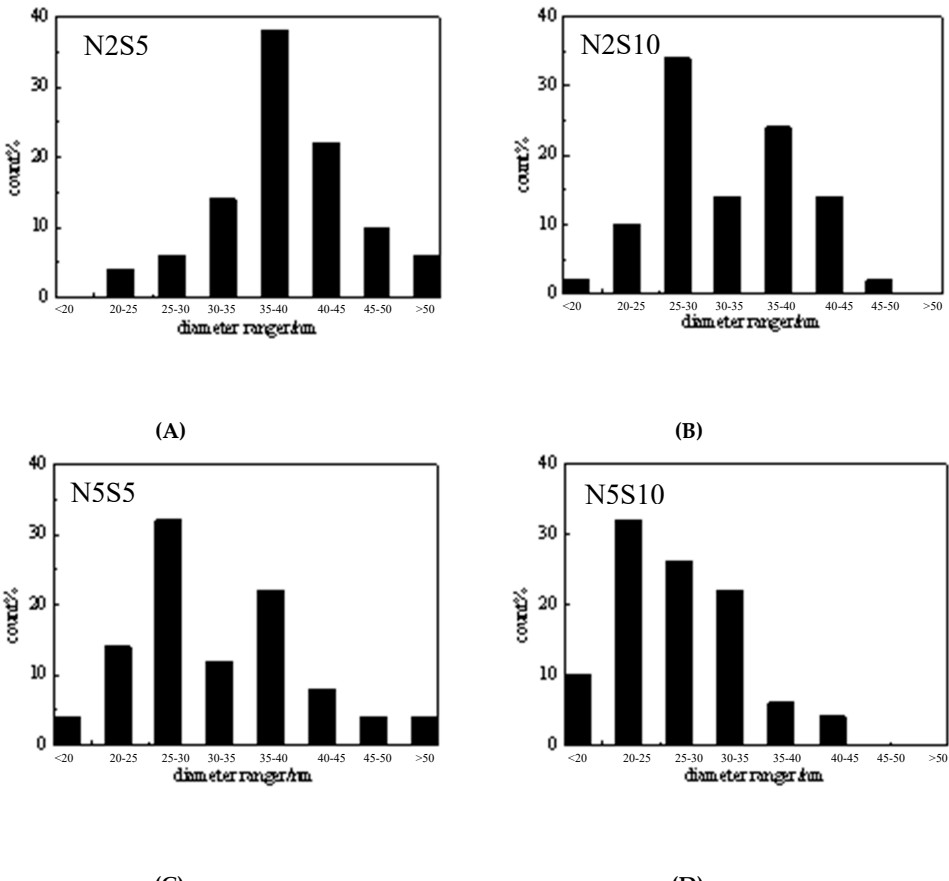

**Figure 6.** Size distribution map of PCF cellulose nanofibers. (N2S5(**A**): 2 wt% NaOH and 5 min sonication, N5S5(**B**): 5 wt% NaOH and 5 min sonication N2S10(**C**): 2 wt% NaOH and 10 min sonication, N5S10(**D**): 5 wt% NaOH and 10 min sonication).

*3.3. Differences in Density and Porosity of Aerogels Prepared by Different Sols*

The density and porosity of nanocellulose aerogels prepared by different sols are shown in Table 1.

**Table 1.** Different sol was prepared nanofiber density and porosity of the aerogel.

| Sample | Density (g/cm$^3$) | Porosity (%) |
|---|---|---|
| 2% NaOH/ultrasound 5 min | 0.0034 | 99.77 |
| 5% NaOH/ultrasound 5 min | 0.0046 | 99.70 |
| 2% NaOH/ultrasound 10 min | 0.0036 | 99.76 |
| 5% NaOH/ultrasound 10 min | 0.0044 | 99.71 |

*3.4. Structural Differences of Aerogels Prepared by Different Sols*

Figure 7 showed the morphology of aerogel under different treatment conditions. After freezing, the sol had almost no shrinkage, and after 5 min of high frequency ultrasound, a well-formed nanocellulose aerogel could be obtained by freeze-drying. The former research work of our group [28] studied the characterizations of PCF and their potential for enzymatic hydrolysis. The sol with a

concentration of 1 wt% was prepared by high frequency ultrasound, and the nanocellulose aerogel was prepared by freeze-drying, with the ultrasonic time for 50 min. This showed the great advantage of low energy consumption when PCF is used in high value field.

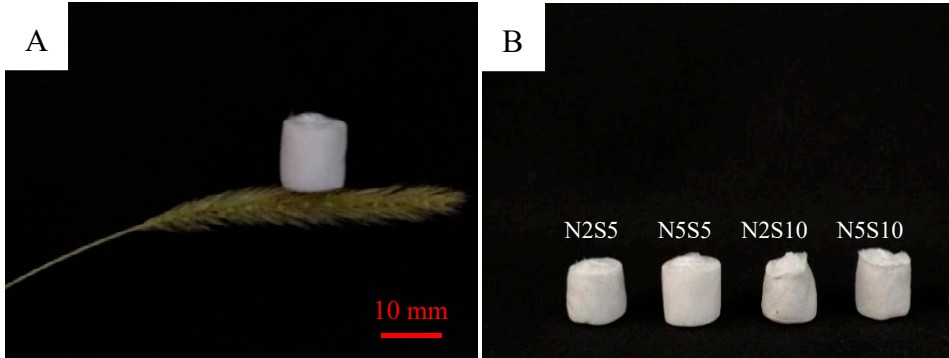

**Figure 7.** (**A**): macro figure of aerogel; (**B**): macro figure of nanocellulose aerogels prepared under; different processing conditions (N2S5: 2 wt% NaOH and 5 min sonication, N5S5: 5 wt% NaOH and 5 min sonication, N2S10: 2 wt% NaOH and 10 min sonication, N5S10: 5 wt% NaOH and 10 min sonication).

Figure 8 is a microscopic appearance diagram of the nanocellulose aerogel prepared under different conditions. It can be seen that nanocellulose aerogel prepared under different conditions can form a porous honeycomb three-dimensional structure, which also explained the ultra-low density and high porosity of aerogels. Additionally, the aerogel consisted of countless small pore-shaped cells with similar shape, each of which was made up of nanocellulose slices and connected to each other. This was due to the decrease of the internal moisture of the nanocellulose sol during the drying process, resulting in the connection of the hydroxyl groups between the adjacent nanocellulose in the solution by the action of hydrogen bonding [31]. In addition, when the liquid nitrogen was frozen, the cold source was outside the sample, there is an outward-to-inward tendency inside the aerogel.

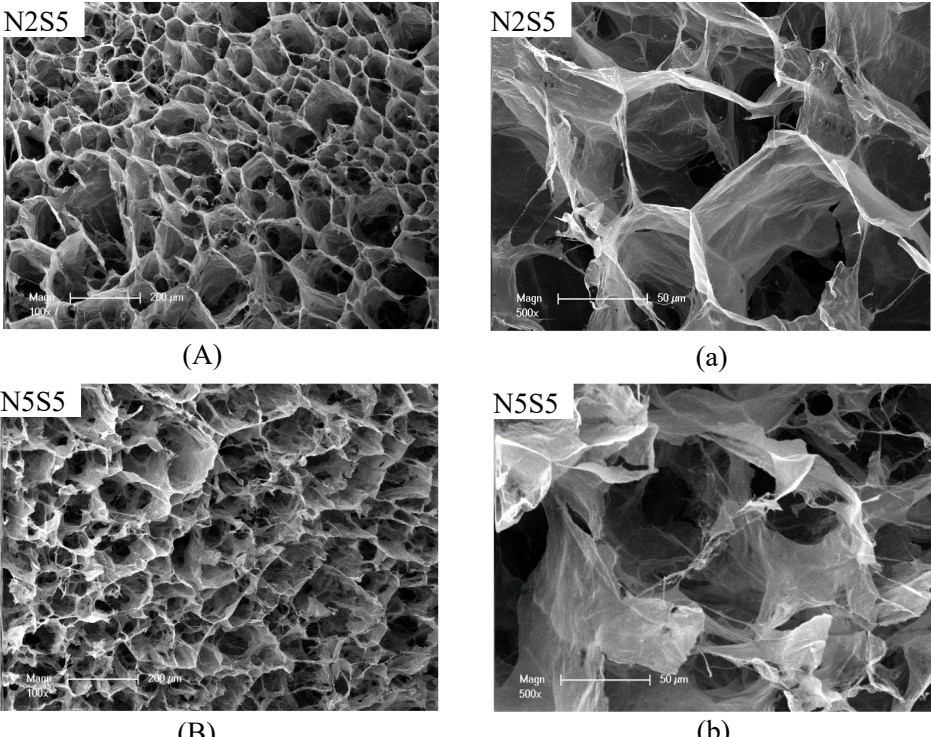

**Figure 8.** *Cont.*

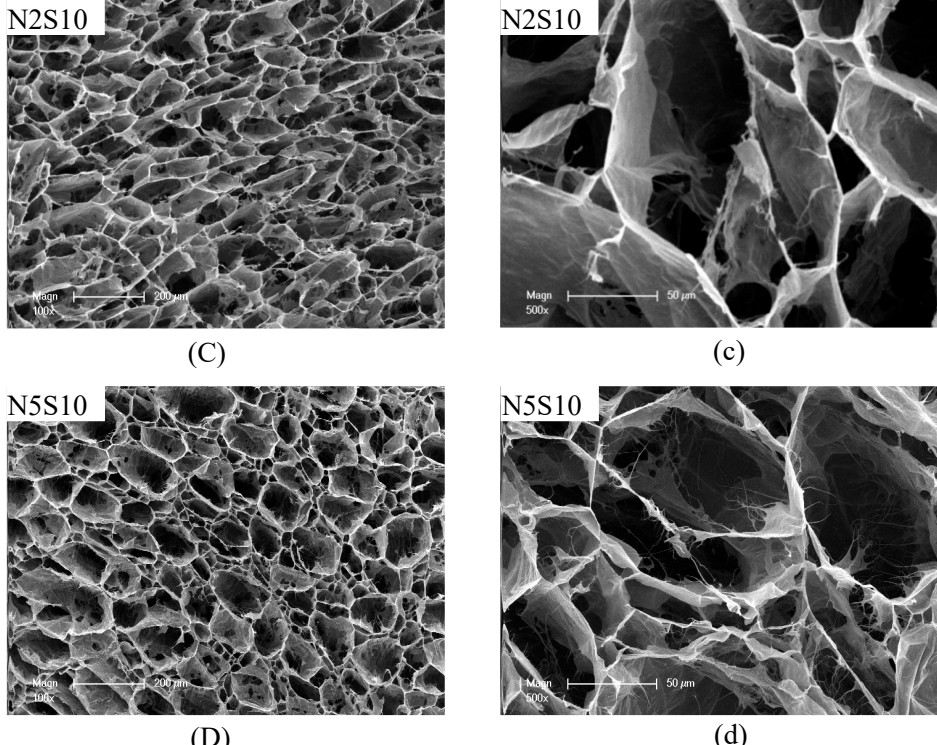

**Figure 8.** N2S5 (**A**): 2 wt% NaOH and 5 min sonication, N5S5(**B**): 5 wt% NaOH and 5 min sonication, N2S10(**C**): 2 wt% NaOH and 10 min sonication, N5S10(**D**): 5 wt% NaOH and 10 min sonication ((**a**)(**b**)(**c**)(**d**) for its corresponding enlargered version).

It can be seen from Figure 8 that the structure of aerogels treated with 2% NaOH solution and 5 min sonication was similar to those treated with 5% NaOH solution and 5 min sonication. The three-dimensional network structure was basically regular, but its internal pore size was different. Additionally, the distribution of pores with different sizes was not uniform, the pore size was about 200 μm for large pore and 50 μm for small pore, which indicated that different concentration of NaOH has little effect on the morphology of aerogel when ultrasonic time was 5 min. The structure of N2S10 was similar to N5S10. The three-dimensional network structure consisted of pores with different sizes; however, the pores tended to be ordered arrangement. After treatment with 5% NaOH solution and 10 min sonication, this phenomenon was more obvious, and the pore of different sizes was arranged alternately. The size of the large pore was 100–200 nm and the size of the small pore was about 40 nm. The above two groups showed that the concentration of NaOH solution and ultrasonic time had a certain influence on the structure of nanocellulose aerogels, but ultrasonic time had a greater effect on the structure of nanocellulose aerogels.

*3.5. Differences in Thermal Stability of Aerogels Prepared by Different Sols*

Figure 9 is a thermogravimetric analysis diagram of aerogels prepared under different conditions. It can be seen from Figure 9A that there was only a small amount of weightlessness in each group of aerogel samples between 0 °C and 200 °C, which was caused by the weightlessness of trace moisture in the sample. The weight loss was mainly between 300 °C and 350 °C, when the cellulose molecular chain was destroyed. However, the difference of pyrolysis temperature among the three groups was small, ranging from 267 °C to 278 °C, indicating that ultrasonic treatment had little effect on thermal stability. From the latter half of the TG curve, it can be seen that nanocellulose aerogels were more thermal decomposed than the cellulose without ultrasound. This was because during the pyrolysis process of aerogels, the hydrogen bonds of the molecular chains were recombined after breaking down,

thus forming a more regular network structure again. Therefore, more heat was needed to decompose these structures [32,33].

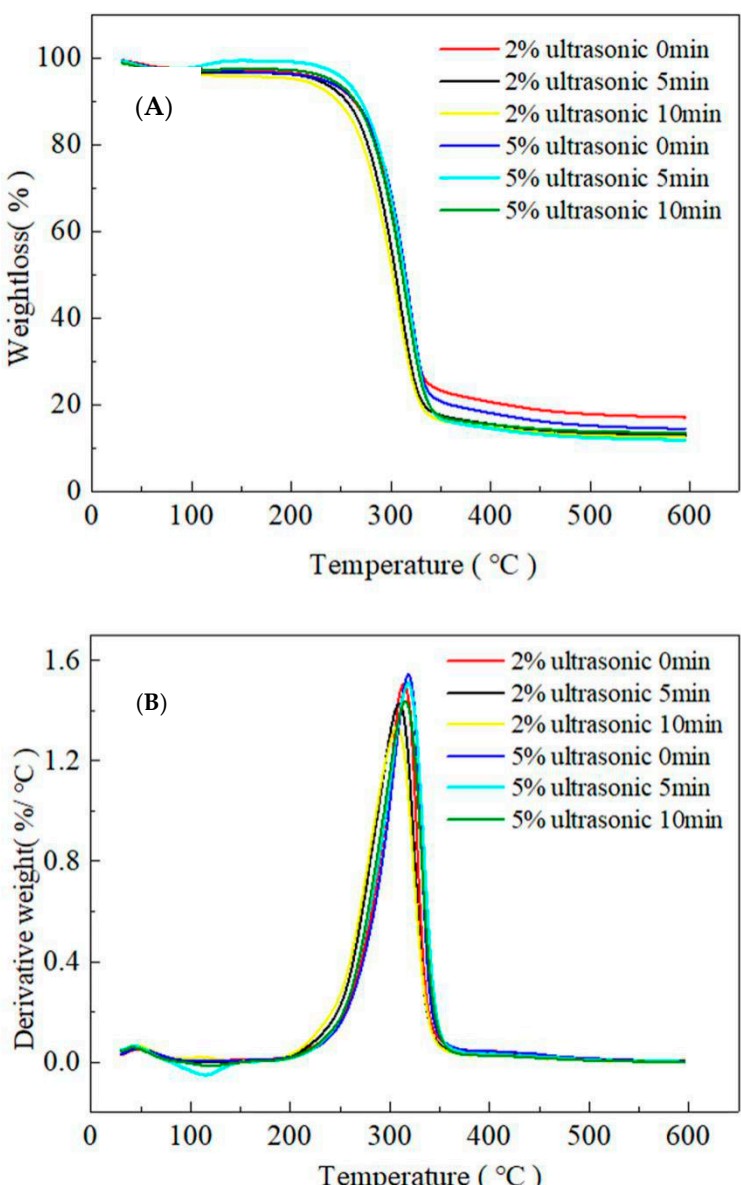

**Figure 9.** Thermogravimetric analyzer (TGA) (**A**) and DTG (**B**) for aerogel prepared at different conditions.

*3.6. Differences in Mechanical Properties of Aerogels Prepared by Different Sols*

Figure 10 shows the stress-strain curves of aerogels prepared under four different conditions. With the increasing of stress, the aerogels had three stages characteristic curves. In the first part, the amount of deformation of aerogels was smaller when the stress was small, thus, the characteristics of elastic deformation appeared. In the second part, with the increasing of the stress, when the yield stress of the material was reached, a gentle deformation region appeared [34]. In the third part, the stress continued to increase, because of the multi-voids structure in aerogels, the deformation became smaller and the structure tended to be compact [35].

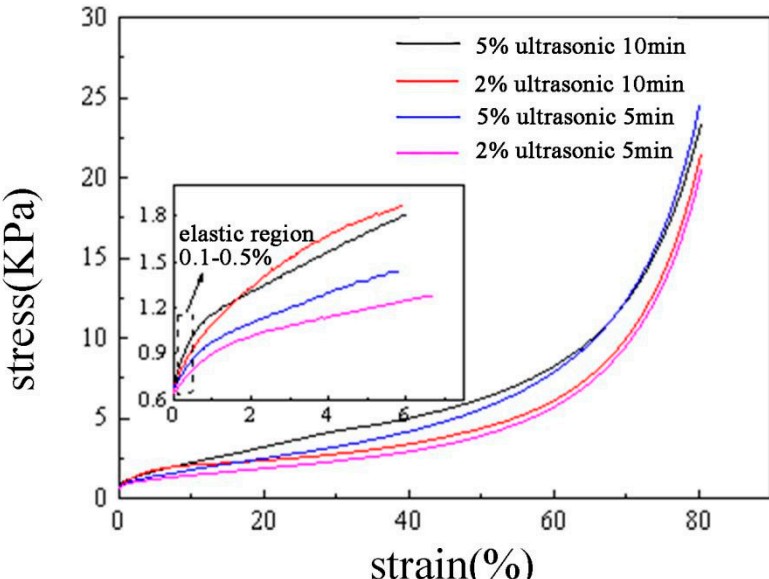

**Figure 10.** Stress-strain curve.

Figure 11 shows the difference in the mechanical properties of the aerogel prepared under different conditions. It can be seen that the range of elastic modulus of the nanocellulose aerogel was 30–52 kPa. With the increase of NaOH solution concentration and ultrasonication time, the mechanical property was enhanced and the elastic modulus of the nanocellulose aerogel was gradually increased, reaching the maximum value of 52 kPa when the PCF being treated with 5 wt% NaOH solution and then 10 min sonication. The aggregation of PCF in the sol will increase due to the fact that sonication can evenly disperse the fiber of PCF for a certain time, which might be result in the increasing of elastic modulus.

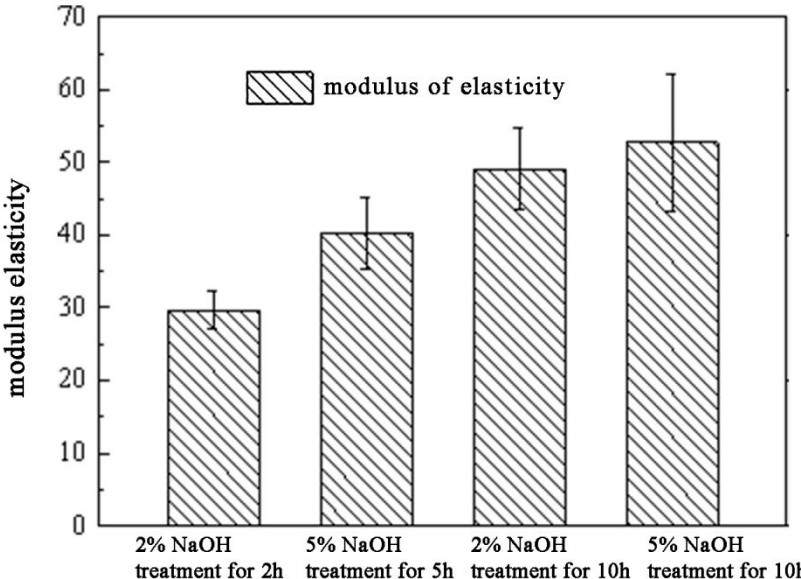

**Figure 11.** Differences in the mechanical properties of the aerogel prepared under different conditions.

## 4. Conclusions

In this study, the microstructure and chemical composition of PCF were systematically explored. The chemical purification and ultrasonic treatment were used to prepare for the PCF nanocellulose. The properties of the nanocellulose were systematically tested. The aerogels were prepared by using the nanocellulose. The elastic modulus of the nanocellulose aerogel was 30-52 kPa and the mechanical

properties increased with the increase of NaOH solution concentration and ultrasonication time. The maximum value reached 52 kPa after the treatment of 5 wt% NaOH solution and 10 min sonication. Therefore, it is feasible to use PCF as raw material to prepare aerogel materials with good properties.

**Author Contributions:** Conceptualization, H.W. and Y.W.; writing—original draft preparation, M.S., X.W. and Y.W.; investigation, X.W.; methodology, T.S. and H.C.; funding acquisition, H.W. and Y.W.

**Funding:** The authors gratefully acknowledgement the financial support from the project funded by the Basic Scientific Research Funds of International Center for Bamboo and Rattan (1632018016), Practical innovation training program for Nanjing Forestry University students (2018NFUSPITP643) and Yihua Lifestyle Technology Co. Ltd. Projects funded (YH-JS-JSKF-201904003).

**Conflicts of Interest:** The authors declare no conflict of interest.

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
