# Peer review of "Preparation of Nanocellulose Aerogel from the Poplar (Populus tomentosa) Catkin Fiber"

_forests, doi:10.3390/f10090749_

Round 1

Reviewer 1 Report

This paper reports a way to extract nanocellulose from poplar catkin fiber (a "biowaste" in a certain part of China), with aerogels as demonstrating application materials. Processing parameters including NaOH concentration and disintegration (sonication) time are found to affect the nanocellulose fibril morphology and width. Moreover, this also has impacts on the nanostructure and mechanical properties of the aerogels. The results are solid, however, the way to present and discuss such results are not well organized. Also, many figures need to be adjusted for publication quality. Thus minor revision is recommended.

Some scientific questions:

1. There are 4 different nanocellulose samples in total:

2%wt NaOH + 5min sonication; 5%wt NaOH + 5min sonication; 2%wt NaOH + 10min sonication; 5%wt NaOH + 5min sonication.

They have been named differently in almost every figure. There are simply so many confusions in the manuscript now:

Figure 5 – A A B B a a b b

Figure 6- A B C D

Figure 7B – a b c d

Figure 8 – A B C D

What is worse: it takes so many sentences to just describe them. For example, line 292 to 297, it takes 5 lines to just tell the sample names…

Please name the sample in a much easier way to make it clear to identify which is which. For example - Nanocellulose(2%+5min) or Nanocellulose-N2S5, or some other ways. Then make a table linking their names to their processing parameter.

2. Table 1 and Figure 8. The author states that the NaOH concentration and sonication affects the aerogel nanostructure. But by looking at Figure 8 A,B,C and D, there are hardly any differences. Actually, macro-size pores in nanocellulose aerogels depend on the density of aerogel (which corresponding to the concentration of nanocellulose dispersion before freeze-drying). So in fact, it will be best if all the aerogels are made into the same density to have a fair comparison. While the nano-size pores in nanocellulose aerogel may depend on the fibril morphology. Please write the discussion part here carefully.

3. Figure 10 and Figure 11. The mechanical properties of aerogels are highly influenced by the density. But the authors ignore such factor in the discussion part. Also, there is no explanation of why the modulus kept increasing with more NaOH treatment and sonication.

4. Do authors have the lignin, hemicellulose, and cellulose contents of the poplar catkin fiber (original, treated)? As well as the final nanocellulose?

Also some minor suggestions:

Line 18: “After the chemical purification of PCF, the nanocellulose was treated with 5 wt% NaOH solution and was ultrasound for 5 min and 10 min, respectively.” It is not “nanocellulose” before the treatment, so “nanocellulose” should be “poplar catkin fibers” here.

Line 33: “northern and northwestern regions.” To “northern and northwestern regions in China”

Line 43: “The 42 microstructure and chemical composition of the PCF are the main determinants of its properties.” Please add REF to support such a statement.

Line 86: “PCF was used as the original material in this experiment because of its high content of cellulose and low content of hemicellulose and lignin.” Any REF? and what are the values?

Figure 1: Figure 1A and Figure 1B has different fond size.

Figure 9: The colors of red/pink and light-blue/dark-blue are too close. Please change.

Line 361: “flocculating fiber nanofibers.” “flocculating”??

Author Response

I am pleased to resubmit for the revised version of manuscript entitled “Preparation of nanocellulose aerogel from the poplar (Populus tomentosa) catkin fiber”. Thank you for reading our manuscript and reviewing it. Those comments are all valuable and very helpful for revising and improving our paper. We have revised our manuscript carefully and have made correction which we hope meet with approval. So we have sent the revised manuscript and have highlighted changes by using the track change mode. The main corrections in the paper and the responds to the reviewers’ comments are as following:

Response to reviewer 1:

There are 4 different nanocellulose samples in total:2%wt NaOH + 5min sonication; 5%wt NaOH + 5min sonication; 2%wt NaOH + 10min sonication; 5%wt NaOH + 10min sonication.They have been named differently in almost every figure. There are simply so many confusions in the manuscript now:

Figure 5 – A A B B a a b b

Figure 6- A B C D

Figure 7B – a b c d

Figure 8 – A B C D

What is worse: it takes so many sentences to just describe them. For example, line 292 to 297, it takes 5 lines to just tell the sample names…Please name the sample in a much easier way to make it clear to identify which is which. For example - Nanocellulose(2%+5min) or Nanocellulose-N2S5, or some other ways. Then make a table linking their names to their processing parameter.

Answer: Line 257 to 270, the sample was named as N5S5 to indicate that it was treated by 5 wt% NaOH and 5 min sonication. And the corresponding treatment of each diagram has also indicated like this: A: 2 wt% NaOH and 5 min sonication (N2S5), B: 5 wt% NaOH and 5 min sonication (N5S5), C: 2 wt% NaOH and 10 min sonication (N2S10), D: 5 wt% NaOH and 10 min sonication (N5S10).

Table 1 and Figure 8. The author states that the NaOH concentration and sonication affects the aerogel nanostructure. But by looking at Figure 8 A,B,C and D, there are hardly any differences. Actually, macro-size pores in nanocellulose aerogels depend on the density of aerogel (which corresponding to the concentration of nanocellulose dispersion before freeze-drying). So in fact, it will be best if all the aerogels are made into the same density to have a fair comparison. While the nano-size pores in nanocellulose aerogel may depend on the fibril morphology. Please write the discussion part here carefully.

Answer: Yes, it`s a very good suggestion. The density control is very important for the properties of aerogel. And in this study the aerogels look similar in structure but their internal pore and the distribution of pore size are slightly different, which suggests that the concentration of NaOH solution and ultrasonic time had a certain influence on the structure of nanocellulose aerogels, but ultrasonic time had a greater effect on the structure of nanocellulose aerogels.

Figure 10 and Figure 11. The mechanical properties of aerogels are highly influenced by the density. But the authors ignore such factor in the discussion part. Also, there is no explanation of why the modulus kept increasing with more NaOH treatment and sonication.

Answer: Yes. The density of aerogels plays an significant role in its mechanical properties. With the increase of NaOH concentration and sonicaiton time, the aggregation of PCF in the sol will increase due to the fact that sonication can evenly disperse the fiber of PCF for a certain time, which might be result in the increasing of elastic modulus.

Do authors have the lignin, hemicellulose, and cellulose contents of the poplar catkin fiber (original, treated)? As well as the final nanocellulose?

Answer: Yes. The content of holocellulose of PCF is 96 %, in which the content of celllose is 44.59 % and hemicellulose is 52.32 %. The content of ligin is 2.92 %. After 0.5 wt% sodium chlorite solution treatment for two hours, the lignin of PCF can be basically removed. The higher the concentration of sodium hydroxide, the longer the ultrasonic time, the better the removal effect of hemicellulose. After chemical purification, the loose structure of PCF made it easy to disintegrate, and nanoscale cellulose can be obtained by five minutes of ultrasonic treatment.

Line 18: “After the chemical purification of PCF, the nanocellulose was treated with 2 and 5 wt% NaOH solution and was ultrasound for 5 min and 10 min, respectively.” It is not “nanocellulose” before the treatment, so “nanocellulose” should be “poplar catkin fibers” here.

Answer: Ok, the sentence was changed. After the chemical purification, the PCF was treated with 2 and 5 wt% NaOH solution and was ultrasound for 5 min and 10 min, respectively.

Line 33: “northern and northwestern regions.” To “northern and northwestern regions in China”

Answer: Thanks. It has been clarified as your suggest.

Line 43: “The 42 microstructure and chemical composition of the PCF are the main determinants of its properties.” Please add REF to support such a statement.

Answer: Ok, the reference was added “Zhang, XX.; Li, ZQ.; Yu Y.; Wang, HK. Characterizations of poplar catkin fibers and their potential for enzymatic hydrolysis. J. Wood Sci. 2018, 1-5, doi:10.1007/s10086-018-1710-3.”.

Line 86: “PCF was used as the original material in this experiment because of its high content of cellulose and low content of hemicellulose and lignin.” Any REF? and what are the values?

Answer: The content of holocellulose of PCF is up to 96 %, in which the content of celllose is 44.59 % and hemicellulose is 52.32 %.The content of ligin is 2.92 %. This was tested based on GB/T 2677.10, GB/T 2677.8 and GB/T 744.

Figure 1: Figure 1A and Figure 1B has different fond size.

Answer: The fond size has been modified to be unified.

Figure 9: The colors of red/pink and light-blue/dark-blue are too close. Please change.

Answer: OK. The colors has been changed to a extremely different color to make it easier to distinguish.

Line 361: “flocculating fiber nanofibers.” “flocculating”??

Answer: It was modified as ‘PCF nanocellulose’.

Reviewer 2 Report

Overview:

In general, the article provides an interesting demonstration of preparing and characterizing cellulose based aerogels from the poplar catkin fiber. Aerogel structure and mechanical response were characterized as a function of % NaOH treatment and sonication time. Overall the work presents the preparation and characterization for a specific source of cellulose via previously demonstrated methods which provides a practical example for it use. The following comments are offered to strengthen the quality of the final manuscript.

General Comments:

Line 33: If specifying geographic, specify the country (assuming China) for a global readership.

Line 54: Is it necessary to distinguish between “domestic and foreign scholars”?

Line 92: BET characterization is indicated but no isotherms are shown in a figure. This should be done if performed.

Line 97: Was the PCF from a commercial vendor? If so, state the vendor and product? If collection, state how the PCF was collected.

Lines 102/108: spell out acronyms for clarity.

Line 114: specify instrument brand/vendor and model number.

Line114-117: Was there temperature control on the sonicator? If so how and what were the settings?

Line 119: What was the weight percent of the nanocellulose sol? This is a critical detail that places the results that follow in context.

Line 123: What is the purpose of the precooling step at “-4 °C”? Was the refrigerator temperature actually -4 °C or +4 °C ?

Line 132: how were samples prepared? What was the substrate?

Line 133: Specify AFM brand and model.

Line 135: Specify probe brand and model.

Line 135: specify AFM mode, ie was tapping mode used?

Line 145: define ρ1 and ρ2 for clarity.

Line 150: specify gold sputter coater brand, model, and coating settings to include time.

Line 154: Specify TGA model.

Line 159: Compression distance is specified as 90 mm for a sample only 13 mm high. This does not seem physically possible.

Figure 1: Recommend adding “a.u.” to units on y-axis; recommend resizing figures to have the same font size for text for panel A and B. Use of “magnifying” in the caption is awkward.

Is the use of the word “grinding” throughout the text used to indicate “sonication”? If so, recommend using “sonication.”

Figure 2: Figure legend – recommend using “control” rather than “check” and using “control” throughout the other figures.

Figure 2 and 3: the use of arrows and wavenumbers in the figures would help clarify text discussion.

Figure 3: recommend detailing A and B in the caption.

Figure 4: the use of “holo” is unclear.

Figure 5: Height and phase scales are needed in AFM images. The use of A, B and a, b as panel designations is confusing. One possible solution is to detail preparation conditions on the AFM images.

Line 235: The use of “crushing” here – is it “grinding”/”sonication”? Again, recommend using sonication throughout the manuscript.

Line 235: What the weight % of the solution during sonication?

Line 249: were all the measurements made of a single sample image? What software was used for measurements?

Table 1: Differences in density and porosity do not seem significant between sample preparation conditions.

Figure 7b: A scale bar would be useful in this photograph.

Line 289: A sol weight percent of 1 % is specified. Was this used in all sample prep? If so specify and add to Methods.

Figure 9: Recommend using “TGA” versus “TG.” Plots seem very similar and not significantly different – why is the final weight loss % different between samples?

Figure 11: caption wording is awkward

Formatting:

Line 35: Recommend “small” rather than “little”

Line 131: Appears to be a “dangling” dash after the period.

Line 265: “nmaccounting” requires a space.

Line 290: too many spaces.

Line 304: Recommend “bonding” versus “bond”

Line 329: recommend “loss” versus “weightlessness”

Author Response

I am pleased to resubmit for the revised version of manuscript entitled “Preparation of nanocellulose aerogel from the poplar (Populus tomentosa) catkin fiber”. Thank you for reading our manuscript and reviewing it. Those comments are all valuable and very helpful for revising and improving our paper. We have revised our manuscript carefully and have made correction which we hope meet with approval. So we have sent the revised manuscript and have highlighted changes by using the track change mode. The main corrections in the paper and the responds to the reviewers’ comments are as following:

Responce to reviewer 2:

Line 33: If specifying geographic, specify the country (assuming China) for a global readership.

Answer: Ok. Its distribution is indeed in China.

Line 54: Is it necessary to distinguish between “domestic and foreign scholars”?

Answer: Thanks. It is not necessary and it has been adjusted.

Line 92: BET characterization is indicated but no isotherms are shown in a figure. This should be done if performed.

Answer: Sorry, it was not what this manuscript talked about and has been modified.

Line 97: Was the PCF from a commercial vendor? If so, state the vendor and product? If collection, state how the PCF was collected.

Answer: The PCF was collected manually, and each poplar tree can harvest an average of 25 kg of PCF.

Lines 102/108: spell out acronyms for clarity.

Answer: OK.The ‘PCF’ has been clarified as photonic crystal fibre and ‘the pH of PCF’ has been made up.

Line 114: specify instrument brand/vendor and model number.

Answer: OK. The brand and model number of the sonication has been added but that of water analyzer was not available.

Line114-117: Was there temperature control on the sonicator? If so how and what were the settings?

Answer: The ice bath was used to control the temperature and prevented to the higher temperature during sonication.

Line 119: What was the weight percent of the nanocellulose sol? This is a critical detail that places the results that follow in context.

Answer: The weight percent of the nanocellulose sol was 1 wt%.

Line 123: What is the purpose of the precooling step at “-4℃”? Was the refrigerator temperature actually -4℃ or +4℃?

Answer: The refrigerator temperature was -4℃, the purpose is to keep it upright in liquid nitrogen for easy observation.

Line 132: how were samples prepared? What was the substrate?

Answer: The method of preparing methods has been added.

Line 133: Specify AFM brand and model.

Answer: It has been specified.

Line 135: Specify probe brand and model.

Answer: The Berkovich probe was used in this study.

Line 135: specify AFM mode, ie was tapping mode used?

Answer: Yes, the tapping mode was used in this study.

Line 145: define ρ1 and ρ2 for clarity.

Answer:ρ1 and ρ2 has been clarified.

Line 150: specify gold sputter coater brand, model, and coating settings to include time.

Answer: The brand of gold sputter was not available. And the specific settings has been described in detail in the manuscript.

Line 154: Specify TGA model.

Answer:The TGA model was ‘5500 type’ and has been specified.

Line 159: Compression distance is specified as 90 mm for a sample only 13 mm high. This does not seem physically possible.

Answer: Sorry,‘90 mm’ was a mistake and it was ‘90 %’, which refers to the compression deformation of the aerogel.

Figure 1: Recommend adding “a.u.” to units on y-axis; recommend resizing figures to have the same font size for text for panel A and B. Use of “magnifying” in the caption is awkward.

Answer: OK. Figure 1 has been modified. The ‘magnifying’ in the caption has been replaced by ‘magnified’.

Is the use of the word “grinding” throughout the text used to indicate “sonication”? If so, recommend using “sonication.”

Answer: Yes. The word ‘grinding’ throughout the text has been replaced by ‘sonication’.

Figure 2: Figure legend – recommend using “control” rather than “check” and using “control” throughout the other figures.

Answer: Thanks. The legend of Figure 2 and Figure 3 has been modified as ‘control’.

Figure 2 and 3: the use of arrows and wavenumbers in the figures would help clarify text discussion.

Answer: Yes. The arrows and local amplification image has been added. It can be concluded from Figure 2 that the nanocellulose prepared by PCF can greatly reduce the amount of chemical reagents and the processing time. And Figure 3 showed that 5 wt% NaOH treatment for 4 h is the most effective preparation methods.

Figure 3: recommend detailing A and B in the caption.

Answer: The meaning of each curve of A and B has been described in detail in the caption.

Figure 4: the use of “holo” is unclear.

Answer: Ok. It has been replaced by ‘cellulose’.

Figure 5: Height and phase scales are needed in AFM images. The use of A, B and a, b as panel designations is confusing. One possible solution is to detail preparation conditions on the AFM images.

Answer: The preparation of A, B and a, b has been described in detail, such as the sample was named as N5S5 to indicate that it was treated by 5 wt% NaOH and 5 min sonication. And the corresponding treatment of each diagram has also indicated like this: A: 2 wt% NaOH and 5 min sonication (N2S5), B: 5 wt% NaOH and 5 min sonication (N5S5), C: 2 wt% NaOH and 10 min sonication (N2S10), D: 5 wt% NaOH and 10 min sonication (N5S10).

Line 235: The use of “crushing” here – is it “grinding”/”sonication”? Again, recommend using sonication throughout the manuscript.

Answer: Thanks. The mistake has been cleared.

Line 235: What the weight % of the solution during sonication?

Answer: The weight % of the solution has been specified. The weight percent of the nanocellulose sol was 1 wt%.

Line 249: were all the measurements made of a single sample image? What software was used for measurements?

Answer: The data was measured from two parallel samples and the software of NanoScope 1.40.0.0 (Bruker, Germany) was used for measurements.

Table 1: Differences in density and porosity do not seem significant between sample preparation conditions.

Answer: Yes. Different conditions has no appearant influence on the density and porosity of the aerogel, however, the compression property was distinct influenced by different conditions.

Figure 7b: A scale bar would be useful in this photograph.

Answer: Figure 7b is the macroscopic morphology figure of aerogel prepared under different conditions, so the scale bar is not necessary.

Line 289: A sol weight percent of 1 % is specified. Was this used in all sample prep? If so specify and add to Methods.

Answer: Yes. The weight percent was added to the methods.

Figure 9: Recommend using “TGA” versus “TG.” Plots seem very similar and not significantly different – why is the final weight loss % different between samples?

Answer: Ok. It is more appropriate to refer ‘the thermogravimetric analysis’ as ‘TGA’. When it comes to the reasons for the difference of final weight loss between samples,my explanation is that different treatment methods led to the changes in the molecular structure of cellulose, resulting in the difference in weight loss.

Figure 11: caption wording is awkward

Answer: Thanks. The caption wording has been clarified. 

Formatting:

Line 35: Recommend “small” rather than “little”

Answer: Ok, it was changed.

Line 131: Appears to be a “dangling” dash after the period.

Answer: Thanks. It was meaningless and has been deleted.

Line 265: “nmaccounting” requires a space.

Answer: OK. The manuscript has been revised as ‘which was at 25~30 nm accounting for 32%’.

Line 290: too many spaces.

Answer: The surplus spaces have been deleted. Thanks.

Line 304: Recommend “bonding” versus “bond”

Answer: Yes. ‘Hydrogen bonding’ is a fixed collocation and the mistake has been modified in the manuscript.

Line 329: recommend “loss” versus “weightlessness”

Answer: Thanks. Obviously, it was more accurate to use the ‘weightlessness’ of trace moisture than the ‘loss’ of it.

Round 2

Reviewer 2 Report

Line 91: Define "GB/T"

Line 125: Should the refrigerator temperature be +4deg C rather than -4deg C?

Figure 7B: A scale bar was added to Figure 7A, but would still be useful for 7B, as well.